# Dissecting Stemness in Aggressive Intracranial Meningiomas: Prognostic Role of SOX2 Expression

**DOI:** 10.3390/ijms231911690

**Published:** 2022-10-02

**Authors:** Rina Di Bonaventura, Maurizio Martini, Tonia Cenci, Valerio Maria Caccavella, Valeria Barresi, Marco Gessi, Alessio Albanese, Liverana Lauretti, Roberto Pallini, Quintino Giorgio D′Alessandris, Alessandro Olivi

**Affiliations:** 1Department of Neurosurgery, Fondazione Policlinico Agostino Gemelli IRCCS, Università Cattolica del Sacro Cuore, 00168 Roma, Italy; 2Department of Pathology, Fondazione Policlinico Agostino Gemelli IRCCS, Università Cattolica del Sacro Cuore, 00168 Roma, Italy; 3Department of Pathology, Università di Verona, 37129 Verona, Italy

**Keywords:** meningioma, SOX2, atypical meningioma, survival, recurrence, progression

## Abstract

Meningiomas are mostly benign tumors that, at times, can behave aggressively, displaying recurrence despite gross-total resection (GTR) and progression to overt malignancy. Such cases represent a clinical challenge, particularly because they are difficult to recognize at first diagnosis. SOX2 (Sex-determining region Y-box2) is a transcription factor with a key role in stem cell maintenance and has been associated with tumorigenesis in a variety of cancers. The purpose of the present work was to dissect the role of SOX2 in predicting the aggressiveness of meningioma. We analyzed progressive/recurrent WHO grade 1–2 meningiomas and WHO grade 3 meningiomas; as controls, non-recurring WHO grade 1 and grade 2 meningioma patients were enrolled. SOX2 expression was evaluated using both immunohistochemistry (IHC) and RT-PCR. The final analysis included 87 patients. IHC was able to reliably assess SOX2 expression, as shown by the good correlation with mRNA levels (Spearman R = 0.0398, *p* = 0.001, AUC 0.87). SOX2 expression was an intrinsic characteristic of any single tumor and did not change following recurrence or progression. Importantly, SOX2 expression at first surgery was strongly related to meningioma clinical behavior, histological grade and risk of recurrence. Finally, survival data suggest a prognostic role of SOX2 expression in the whole series, both for overall and for recurrence-free survival (*p* < 0.0001 and *p* = 0.0001, respectively). Thus, SOX2 assessment could be of great help to clinicians in informing adjuvant treatments during follow-up.

## 1. Introduction

Intracranial meningiomas are the most common primary brain tumors, accounting for one-third of all cases in adults. Typically, they are benign tumors curable by surgery; however, even after seemingly complete resection, some cases can recur. Recurrence affects up to 18% of patients in the first 5 years after resection and as many as 25% within 10 years. WHO grade 1 meningiomas recur in 7 to 25% of cases [1], and a subset of grade 1 meningiomas can de-differentiate and progress to higher grade. WHO grade 2 and 3 meningiomas show a higher incidence of recurrence (29–52% and 50–94% at 5–10 years, respectively). WHO grade 3 meningiomas (anaplastic meningiomas) could originate *de novo* or from the progression of lower grade meningiomas (progressive anaplastic meningiomas). To ensure adequate treatment, there is an urgent need to predict early the clinical aggressiveness of meningioma. Although histopathology remains the gold standard for grading [2], prediction of tumor recurrence and management of aggressive meningiomas remain major challenges in neuro-oncology [3]. In particular, grade 2 meningiomas represent a clinically heterogeneous group of tumors, among which some behave more favorably, while others recur even after gross total resection, sometimes progressing toward malignant forms. Reportedly, even grade 1 meningiomas display marked genetic and epigenetic variability. Many genetic, cytogenetic and epigenetic alterations have been described in meningioma, and their diagnostic and prognostic role is currently a matter of intensive investigation [4]. Over the past few decades, substantial evidence has convincingly revealed the existence of cancer stem cells (CSCs) in meningioma [5,6,7]. CSCs are functionally defined by their abilities to self-renew, differentiate and phenocopy the original tumor when xenotransplanted in immunocompromised rodents [8]. The CSC population contributes to tumor progression, therapeutic resistance and disease recurrence. Among the CSC markers, the Sex-determining region Y (SRY)-box (SOX) factors are a family of transcriptional regulators that carry out crucial functions during embryonic development [9,10,11]. Previous experiments of our group in mouse high-grade gliomas identified a network of critical tumor-suppressive Sox2 targets, whose inhibition is involved in glioma CSC maintenance, therefore defining new therapeutic targets [12]. Herein, we evaluated the expression of SOX2 in a series of recurrent and progressive meningiomas in order to assess its prognostic role.

## 2. Results

### 2.1. Clinical Features and Subgrouping of Patients

Overall, we included 87 patients who underwent 148 operations (Table 1). At first diagnosis, the WHO grade was grade 1 in 27 patients, grade 2 in 44 patients, and grade 3 in 16 patients. The median follow-up was 122.9 months. According to the outcome, the patients were categorized into six groups: Group 1B, WHO grade 1 meningiomas, which did not recur within 10 years (20 patients); Group 2B, WHO grade 2 meningiomas, which did not recur within 5 years (19 patients); Group 3DN, WHO grade 3 de novo meningiomas (16 patients); Group 1P, WHO grade 1 meningiomas, which recurred and progressed to higher grade (7 patients); Group 2P, WHO grade 2 meningiomas that recurred and progressed to higher grade (13 patients); Group 2R, WHO grade 2 meningioma, which recurred without progression (12 patients).

Patients in Group 3DN were significantly older compared to Groups 1B, 2B and 2P (*p* = 0.0013, *p* = 0.0122 and *p* = 0.0225, respectively; unpaired *t*-test). Female/male ratio was significantly lower in Group 2P compared to Group 1B (*p* = 0.0324, Fisher exact test). No significant differences regarding tumor location were found.

### 2.2. Validation of SOX2 IHC by RT-PCR

One hundred and eleven meningioma samples were available for SOX2 immunohistochemistry (IHC). Samples at first diagnosis and samples at recurrence were available in all cases. SOX2 immunoreactivity was mainly nuclear, although outbreaks of cytoplasmic staining were not infrequent (Figure 1), as already described [13]. Confocal immunofluorescence using meningioma markers confirmed that the SOX2 signal resides in the nucleus (Appendix A). In order to validate SOX2 IHC [14,15], in a subset of specimens (71 samples), mRNA levels of SOX2 were semiquantitatively assessed by RT-qPCR. SOX2 grading by IHC and mRNA levels were significantly correlated (R = 0.4, *p* = 0.0006, Spearman′s correlation; Appendix A). ROC analysis confirmed the accuracy of SOX2 mRNA evaluation in predicting IHC, with an AUC of 0.8629 (*p* = 0.0001; Appendix A). Namely, SOX2 mRNA levels were remarkably lower in the IHC negative (0) meningioma as compared with SOX2 IHC positive ones (relative medians 0.0350 [0.0085; 0.0710] vs 0.2400 [0.1018; 0.4665], *p* < 0.001, Mann–Whitney U test) (Appendix A).

### 2.3. SOX2 Immunostaining and Meningioma Grade

In the whole series, the fraction of SOX2-positive tumors on IHC was significantly correlated with the meningioma grade. Considering all samples, among grade 1 meningioma, 39.3% showed immunostaining for SOX2; among grade 2 meningioma, 72.9% were positive; among grade 3 meningioma, 94.3% were positive on IHC for SOX2 (*p* < 0.0001, Chi square test; Figure 2A). Immunohistochemistry of samples at the first diagnosis gave similar results (Figure 2B).

We then compared SOX2 expression at the first diagnosis in the six groups, as previously defined. This analysis was designed to investigate whether SOX2 expression might predict the behavior of the tumor. The results are shown in Figure 3. Among grade 1 meningiomas, we found that the percentage of SOX2-positive tumors was significantly higher in Group 1P than in Group 1B, i.e., in progressive vs non-progressive grade 1 meningiomas (87.5 vs 20%, *p* = 0.0042, Fisher exact test). Among grade 2 meningioma, Group 2P, i.e., grade 2 meningioma undergoing anaplastic transformation, had a significantly higher percentage of SOX2-positive cases than Groups 2B and 2R (100 vs 47.4% and 100 vs 64.3%; *p* = 0.0016 and *p* = 0.0391, respectively; Fisher exact test). No significant differences were found between Group 2B and Group 2R. These results suggest that SOX2 expression is an innate feature of meningioma that is intrinsically correlated with aggressive potential since the very first diagnosis. To further confirm this evidence, we compared SOX2 expression at first diagnosis and at recurrence in Group 1P and Group 2P, i.e., in cases in which recurrence was associated with grade progression and found no significant differences (*p* = 0.5312 and *p* = 0.1250, Wilcoxon signed-rank test).

### 2.4. SOX2 Expression Correlates with PFS, OS and Recurrence Risk

Further, we assessed the clinical value of SOX2 expression in predicting the PFS and OS of meningiomas. In the whole series, patients with SOX2-positive meningiomas had PFS and OS significantly lower than those with SOX2-negative meningiomas (median PFS 38.4 months in SOX2-positive meningiomas vs not reached in SOX2-negative tumors; *p* < 0.0001, log-rank test; median OS 173.9 months in SOX2-positive vs not reached in SOX2-negative tumors; *p* = 0.0001 Log-rank test; Figure 4A,B). Moreover, SOX2 expression at first diagnosis was tightly related to the risk of surgical recurrence (*p* < 0.0001, Fisher exact test; Figure 4C).

The negative prognostic value of SOX2 expression was confirmed in patients with grade 1 meningioma (median PFS 101.1 months in SOX2-positive vs not reached in SOX2-negative tumors; *p* = 0.0012, Log-rank test; median OS 175.6 months in SOX2-positive vs not reached in SOX2-negative tumors; *p* = 0.0054, Log-Rank test; n = 27; Figure 5A,B) and in patients with grade 2 meningioma (median PFS 29.0 months in SOX2-positive vs not reached in SOX2-negative cases; *p* = 0.0055, Log-rank test; median OS 173.9 months in SOX2-positive vs not reached in SOX2-negative cases; *p* = 0.0342, Log-rank test; n = 44; Figure 5C,D).

To summarize, SOX2 was shown to be a negative prognostic factor for recurrence and survival in meningioma patients. This notion may be of clinical value in cases of grade 1 meningiomas expressing SOX2, which require special attention, given the risk of recurrence and pathological transition with progression.

## 3. Discussion

The main findings of the present work can be summarized as follows: (i) the expression of the stemness marker SOX2 can be reliably assessed in meningioma by IHC, as this analysis closely reflects the level of SOX2 mRNA; (ii) SOX2 expression in meningioma is a biomarker of tumor aggressiveness; (iii) SOX2 expression is an intrinsic feature of meningiomas since first diagnosis without change through recurrences.

Meningiomas show remarkably wide biologic and histological heterogeneity, which is poorly captured by the current WHO classification. About 20% of these tumors recur and become life-threatening lesions. Atypical and anaplastic meningiomas constitute the most clinically aggressive forms; however, up to 20% of histologically benign meningiomas may also be clinically aggressive. Therefore, there is a desperate need to identify biomarkers able to point out clinically aggressive cases in advance and to refine the current WHO grading system [16]. Some studies showed that the presence of a complex karyotype in benign tumor heralded progression [17], raising the hypothesis that these tumors were intrinsically malignant. However, cytogenetics has not entered the diagnostic routine as the analysis of complex karyotypes may be difficult. Molecular analyses have shown that meningiomas can be widely divided into NF2-mutated and non-NF2-mutated, but the clinical implications of this finding are unclear [4]. Among the various molecular players that have been investigated, TERT has been proposed as a useful tool for meningioma grading; in fact, the rate of telomerase hyperactivity increases in higher grades. However, the heterogeneity of mechanisms of increase in TERT activity in meningiomas is the main limitation to its use as a prognostic marker [18]. Methylome analysis carries a promising role in predicting tumor aggressiveness [19]. Recently, several authors have provided evidence of the existence of a subpopulation of CSCs in meningiomas [6,7,20]. In our study, we investigated the expression and prognostic role of the stem cell marker SOX2. SOX2 is a member of the SOXB1 group, which is required for the maintenance of the embryo before implantation and plays a role in cell fate and in maintaining the identity of progenitors during embryogenesis. It is also important for homeostasis and tissue regeneration by maintaining the activity of stem cells in different compartments, particularly in the central nervous system [21]. The gene encoding SOX2 is located on the long arm of chromosome 3 and the increase of its expression has been correlated with growth, tumorigenesis, chemo-resistance and metastatic capacity in at least 25 different types of cancer, including ovary, lung, skin, brain, breast, prostate, pancreas and, importantly, brain tumors [9,22]. SOX2 controls several characteristics of cancer cells, such as proliferation, epithelium-mesenchymal transition, migration, invasion, metastasis, spherical and colony formation, tumor initiation, and cancer stem cell formation, as well as resistance to apoptosis and chemotherapy [9]. Its relevance to cancer progression is mediated by a very complex signaling network. Four main signaling pathways are involved in SOX2 expression favoring tumor maintenance: TGF-β [23], SHH pathway [24], EGFRvIII [25], RhoA-dependent pathway and focal adhesion kinase (FAK) signaling [26]. Evidence of the role of SOX2 in meningioma is scarce. The very first description comes from a work by Xiao et al., who suggested that the stem cell markers nestin, CD133 and SOX2 may be correlated with the pathological grade.

The data gathered here confirm and expand this suggestion. SOX2 expression was indeed strictly associated with meningioma grade. High grades were populated by substantial proportions of SOX2-positive tumor cells. In addition, our study first pointed out that SOX2 levels correlate with prognosis in terms of PFS and OS, although the design of the study prompts caution on this latter conclusion.

The expression of SOX2 could distinguish those grade 1–2 meningiomas that have a favorable clinical course from those that behave aggressively, irrespective of their WHO grade. Importantly, such differentiation would have been possible since the first diagnosis. This is of particular value because SOX2-positive tumors, which are otherwise histologically benign tumors, should be tightly followed up. On the other hand, in cases of incomplete resection, SOX2 positivity could be an argument in favor of early adjuvant radiotherapy. Notably, similar to other tumors [27,28,29] we showed that SOX2 status can be reliably assessed using immunohistochemistry, which is a cheaper and more easily available technique than semiquantitative RT-PCR, thus reinforcing the clinical value of our findings.

In perspective, SOX2 could also be a therapeutic target. Since direct SOX2 inhibition is unfeasible due to inacceptable toxicity, the main pathways regulated by SOX2 in meningioma could be dissected and their inhibition explored [23,24,30]. This approach could fill the dramatic lack of effective adjuvant treatments for aggressive meningiomas; however, it would require extensive preclinical work before it could be translated into a clinical setting.

Notwithstanding the strong negative prognostic value of SOX2 expression, which has been widely addressed in this study, a small but non-negligible proportion of WHO grade 1, SOX2-positive meningiomas (20%) will eventually behave in a benign way and will not undergo recurrence or progression. This evidence warrants further study. One possible explanation involves the existence of different CSC subtypes that are able to confer a more favorable prognosis to some tumors, as already demonstrated in gliomas [31].

One strong point of the present work is the enrollment of a large series of patients homogeneously treated at a leading brain tumor referral center. Another strong point is the use of 2 different techniques for SOX2 evaluation: immunohistochemistry and semiquantitative RT-PCR. Among the limitations is the retrospective nature of the study, which inevitably reduces the strength of gathered data; however, the large effect size of SOX2 expression in hampering patients′ prognosis increases the overall quality of data and reassures on their generalizability.

To conclude, meningioma represents an extraordinary challenge in neuro-oncology, because, although in most cases it behaves as a benign tumor, a definite subgroup of patients will undergo unexpected recurrence or even neoplastic transition and progression. The current histopathological classification does not seem to predict this risk in a reliable fashion. SOX2 assessment is a promising, powerful molecular tool to accurately stratify the risk of recurrence and progression of disease since the very first diagnosis of meningioma, irrespective of WHO grade. Future perspectives of our work move on dual tracing, namely validation of molecular data on wider series and exploring new therapeutic targets. The validation of the negative prognostic value of SOX2 in meningiomas might also prompt the development of targeted therapies that are able to inhibit the SOX2 pathway.

## 4. Materials and Methods

### 4.1. Patient Enrollment

Between 2004 and 2019, a total of 955 patients were operated on for intracranial meningioma at Fondazione Policlinico Universitario “A. Gemelli” IRCCS, Rome, Italy. Of these, 775 (81.1%) were WHO grade 1, 164 (17.2%) were grade 2, and 16 (1.7%) were grade 3 meningiomas. Among the grossly total resected grade 1 meningiomas, 14 (1.8%) showed recurrence with a transition to higher grade. Among the grossly total resected grade 2 meningiomas, 25 (15.2%) and 16 (9.8%) showed recurrence without transition and with transition to grade 3, respectively. Patients with gross total resection, grade 1 and grade 2 meningiomas showing recurrence with transition to higher grade (up to progressive anaplastic meningioma), grade 2 meningiomas showing recurrence without transition, and grade 3 meningiomas were included in the present study. As controls, we considered patients harboring grade 1 meningioma without tumor recurrence at 10 years after surgery and grade 2 meningiomas without tumor recurrence at 5 years after gross total resection. The study protocol was approved by the Ethics Committee of Fondazione Policlinico Gemelli IRCCS (Prot. ID 3459). All patients signed an informed consent form to use their clinical data and pathological samples.

### 4.2. Data Collection and Follow-Up

Clinical and follow-up data were collected through clinical and pathological reports. Follow-up visits were performed whenever feasible for study purposes. Public registries were checked for the assessment of overall survival (OS). Survival data were censored in November 2021. Disease progression was evaluated using the Response Assessment in Neuro-Oncology Working Group’s proposed criteria for meningiomas [32]. Progression-free survival (PFS) was defined as the time interval between the first diagnosis of meningioma and tumor recurrence or growth of residual tumor requiring treatment. Overall, survival was defined as the timeframe between the first meningioma diagnosis and death from any cause.

### 4.3. Immunohistochemistry for SOX2

Immunohistochemical analysis was performed on 3-μm thick formalin-fixed, paraffin-embedded tissue sections. A rabbit polyclonal anti-SOX2 antibody (1:1000 dilution, Abcam, Cambridge, United Kingdom) was used. Sections were deparaffinized with three xylene washes of 10 min duration each, rehydrated in three ethanol series at decreasing concentrations (5 min at 100%, 5 min at 85%, and 5 min at 70%). Then, the sections were washed with distilled water. For antigen unmasking, paraffin sections were microwave-treated in 0.01M citric acid buffer, pH 6.0 (2 cycles of 5 min each at 750 W). After multiple washes in double-distilled water and PBS, endogenous peroxidases were inhibited with 3% H2O2 for 5 min, followed by further washes in double-distilled water and PBS. Sections were then incubated with primary antibodies for 30 min and subsequently incubated with secondary antibodies (Dako REAL EnVision HRP rabbit/mouse, ENV) for 30 additional min. 3,3′-diaminobenzidine was used as the enzyme substrate to observe specific antibody localization, and Mayer hematoxylin was used as a nuclear counterstain. The sections were dehydrated by immersion in solutions at progressively increasing concentrations of ethanol and then in xylene. Finally, the histological sections were mounted in Canada balsam as permanent slides. Immunostaining was evaluated independently by 3 observers, who were blinded to the patients’ characteristics and survival. Cases with disagreement were discussed using a multiheaded microscope until agreement was achieved. To assess differences in immunoreactivity, the following scoring system was applied: 0, when less than 25% of cells in the tumor specimen showed nuclear or nuclear-cytoplasmic expression of SOX2; +, when more than 25% of cells in the tumor specimen showed nuclear or nuclear-cytoplasmic expression of SOX2.

### 4.4. mRNA Extraction and Semiquantitative Real Time (RT)-PCR for SOX2

After being deparaffined with xylene and rehydrated with ethanol, three 10-µm slides were obtained for each tumor sample. RNA was extracted with the RNeasy FFPE Kit (Qiagen), following the manufacturer′s protocol. The quantity and quality of the RNA were assessed spectrophotometrically (Qiaxpert, Qiagen), evaluating the absorbance at 260 nm. The RT-qPCR reaction for the amplification and quantification of SOX2 mRNA and actin mRNA, which were taken as the internal reference gene, was performed using the CFX96 optical reaction module (Biorad). The sequences of the primers were as follows:

for SOX2, forward 5′-TACAGCATGTCCTACTCGCAG-3′, reverse 5′-GAGGAAGAGGTAACCACAGGG-3′,

for actin, forward 5′-GGCGGCACCACCATGTACCCT-3′, reverse 5′-AGGGGCCGGACTCGTCATACT-3′.

For each sample, 20 µL reaction volume contained 5 ng of RNA, 10 µL of Master Mix SyGreen 1-step (PCRBIOSYSTEMS), 1 µL of 10 mM of each primer and 1 µL of RT enzyme. Thermocycler conditions were as follows: incubation at 45 °C for 10 min to retrotranscribe mRNA to cDNA and denaturation at 95 °C for 2 min, followed by 40 cycles at 95 °C for 15 min and at 60 °C for 30 min. Melting curves were assessed to determine the amplification of each product. The relative levels of the expression of SOX2, normalized by actin, were calculated according to the 2-ΔCq method.

### 4.5. Statistical Analysis

The distribution of continuous variables was assessed by the D′Agostino-Pearson test and Shapiro-Wilk test and then described as normal using mean and standard deviation, if not as median and interquartile range. Categorical variables were described using absolute and relative frequencies and were analyzed via the Chi-square test, using Fisher’s exact test when appropriate. The correlation between ordinal variables was assessed using the Spearman correlation coefficient. Comparison of continuous variables between groups was performed using the unpaired *t*-test for normally distributed variables, otherwise using the Mann–Whitney *U* test. Comparison of continuous variables at different timepoints in the same group was performed using the Wilcoxon Signed Rank test. Survival data were analyzed by building Kaplan–Meier curves, and differences between groups were assessed using the log-rank test. A ROC curve was built to assess the accuracy of RT-PCR in predicting SOX2 positivity, as assessed by immunohistochemistry. A *p* value of less than 0.05 was considered statistically significant. Analyses were performed using Prism 9 (Graph Pad Software, San Diego, CA, USA). This report was drafted based on STARD guidelines.

## Figures and Tables

**Figure 1 ijms-23-11690-f001:**
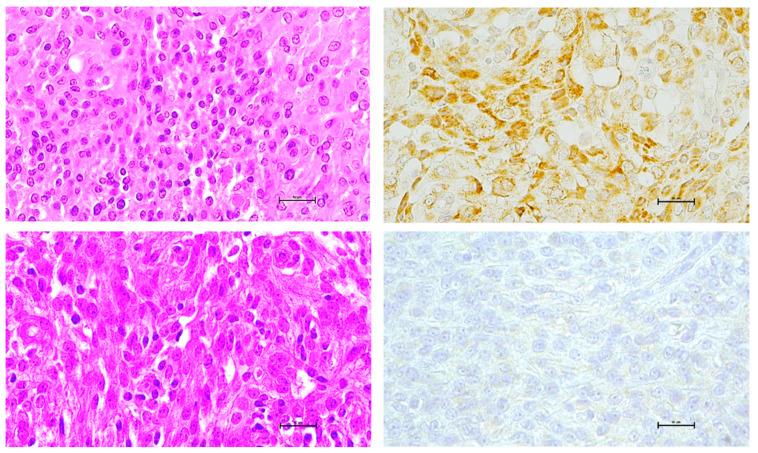
Immunohistochemical evaluation of SOX2. Upper row. WHO grade 3 meningioma, hematoxylin and eosin (H&E) staining (*left)* and strong (3+) SOX2 IHC positivity (*right)*. *Lower row.* WHO grade 1 meningioma, H&E staining *(left)* and SOX2 IHC negativity (*right)*. Scale bar, 50 μm.

**Figure 2 ijms-23-11690-f002:**
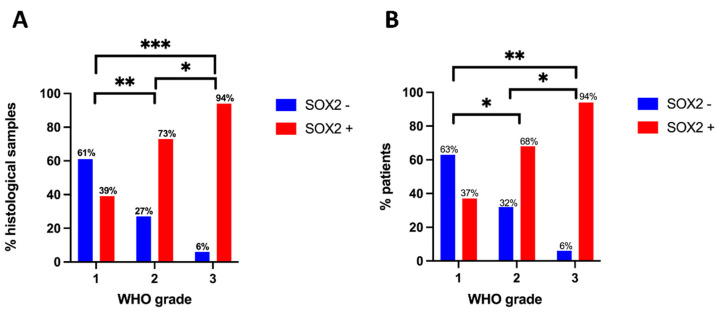
Correlation between SOX ICH status and meningioma grade. (A), Histogram showing analysis performed on all available samples. SOX2-positive cases are significantly higher among WHO grade 3 tumors than among grade 1 (*p* < 0.0001, Fisher′s exact test) and grade 2 meningiomas (*p* = 0.0189, Fisher’s exact test); moreover, SOX2-positive cases are higher in grade 2 than in grade 1 tumors (*p* = 0.0069, Fisher′s exact test). (**B**), Histogram showing analysis performed on samples obtained at the first diagnosis in the whole series. Similar to the analysis shown in (**A**), we found a significantly higher percentage of SOX2-positive cases in grade 3 meningioma compared to grade 1 and 2, and in grade 2 compared to grade 1 meningioma (*p* = 0.0003, *p* = 0.0499 and *p* = 0.0141, respectively; Fisher′s exact test). ** p* < 0.05; ** *p* < 0.01; *** *p* < 0.0001.

**Figure 3 ijms-23-11690-f003:**
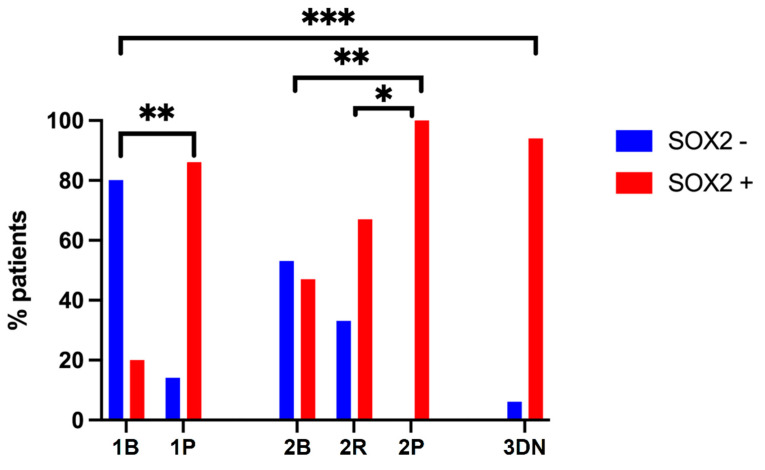
SOX2 IHC expression at first diagnosis in meningioma belonging to different groups. ** p* < 0.05; ** *p* < 0.01; *** *p* < 0.0001.

**Figure 4 ijms-23-11690-f004:**
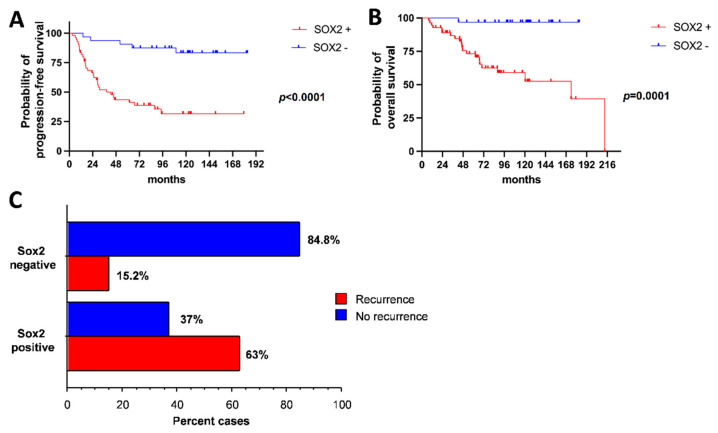
(**A**,**B**) Kaplan–Meier survival curves for PFS (**A**) and OS (**B**) in the whole series. (**C**) Probability of tumor recurrence depending on SOX2 IHC status.

**Figure 5 ijms-23-11690-f005:**
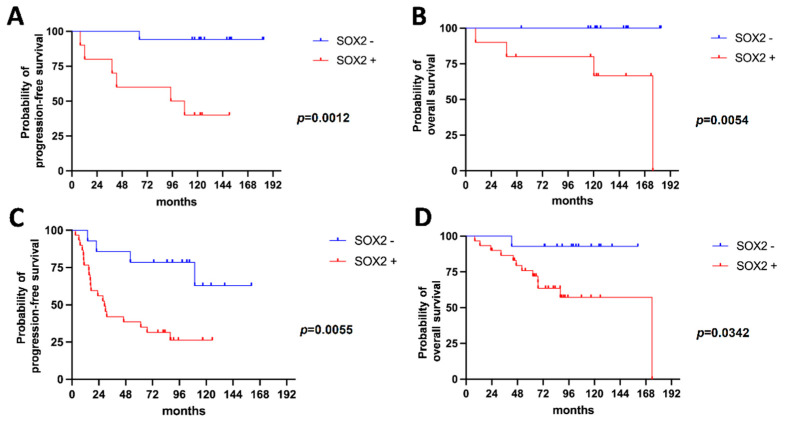
Kaplan–Meier survival curves for PFS (**A**,**C**) and OS (**B**,**D**) in grade 1 meningiomas (**A**,**B**) and in grade 2 meningiomas (**C**,**D**).

**Table 1 ijms-23-11690-t001:** Clinical characteristics of meningioma patients.

Parameter	All Patients	Group					
		1B	1P	2B	2R	2P	3DN
*N*	87	20	7	19	12	13	16
F:M	42:45 0.9	13:7 1.9	3:4 0.8	11:8 1.4	4:8 0.5	3:10 0.3	8:8 1
Age at diagnosis (years)	61.0 ± 11.5	57.8 ± 8.9	57.3 ± 16.9	59.0 ± 11.9	61.4 ± 10.7	60.2 ± 9.1	69.4 ± 11.1
N surgeries, median (range)	1 (1–6)	1	2 (2–6)	1	3 (2–5)	2 (2–6)	1 (1–3)
Follow-up (months)	87.5 ± 47.9	136.0 ± 21.3	88.1 ± 67.9	102.0 ± 22.8	73.8 ± 28.1	57.5 ± 43.5	43.8 ± 40.9

## Data Availability

Source data are available from the Corresponding Author upon reasonable request.

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
