# Peer review of "Dissecting Stemness in Aggressive Intracranial Meningiomas: Prognostic Role of SOX2 Expression"

_ijms, 2022, doi:10.3390/ijms231911690_

Round 1
Reviewer 1 Report
This study explores the impact of SOX2 expression on the prognosis of operated meningiomas.
It takes part in both the major consideration of biological markers in the classification of brain tumors and on the more recent field of study on the specific role of SOX2 in carcinogenesis.
The authors present here a study which seems to be a retrospective case/control study of patients of all grades with or without recurrence after surgery. This type of study allows certain statistical analyzes but fits very poorly into a study focusing on the impact on overall survival or recurrence-free survival.
Thus, on the significant results declared in the discussion (page6, line 168 to 173):
The first one corresponds to the satisfactory association between SOX2 expression in IHC and in mRNA. The data presented does indeed support this statement.
The second, which consists in demonstrating the impact of the SOX2 expression on overall survival and progression free survival just cannot be affirmed in the context of a case/control groups because of the obvious selection bias. To make such a statement, it would have been necessary to follow an unbiased population such as all the patients treated for their grade 1 meningioma, evaluate the expression of SOX2 and assess their PFS and OS. On highly selected patients as the ones included in this study on a case/control manner, the conclusions on the survival data presented must be largely modified and understated.
The last one, which assessed the stability of SOX2 expression from the first surgery to the recurrence, seems to be adequately supported by the experimental datas.
Finally, in the introduction and discussion, it would have been relevant to include reflection on the other molecular analyzes that participate or could participate in the classification of meningiomas.
Overall an interesting and certainly relevant subject but the statements on the survival impact on SOX2 should be understated.
Author Response
Response to Reviewers’ comments
Reviewer #1
This study explores the impact of SOX2 expression on the prognosis of operated meningiomas.
It takes part in both the major consideration of biological markers in the classification of brain tumors and on the more recent field of study on the specific role of SOX2 in carcinogenesis.
The authors present here a study which seems to be a retrospective case/control study of patients of all grades with or without recurrence after surgery. This type of study allows certain statistical analyzes but fits very poorly into a study focusing on the impact on overall survival or recurrence-free survival.
Response: The Reviewer is right on the study design and its implications. As detailed in the following responses, we did our best to deal with it.
Thus, on the significant results declared in the discussion (page6, line 168 to 173):
The first one corresponds to the satisfactory association between SOX2 expression in IHC and in mRNA. The data presented does indeed support this statement.
The second, which consists in demonstrating the impact of the SOX2 expression on overall survival and progression free survival just cannot be affirmed in the context of a case/control groups because of the obvious selection bias. To make such a statement, it would have been necessary to follow an unbiased population such as all the patients treated for their grade 1 meningioma, evaluate the expression of SOX2 and assess their PFS and OS. On highly selected patients as the ones included in this study on a case/control manner, the conclusions on the survival data presented must be largely modified and understated.
Response: as noticed before, we agree with the Reviewer on the concerns over the study design. On the other hand, the study enrolls a non-negligible number of patients and widely shows that SOX2 expression is correlated with the histological grade, the clinical behavior and the recurrence risk of meningioma, thus portending a worsened prognosis of SOX2-positive meningiomas . To deal with the Reviewer’s concern, we have: i) widely revised the abstract; ii) removed the reference to OS and PFS from the first paragraph of the Discussion (page 6, lines 171-172); iii) added in the Discussion the sentence: “though the design of the study prompts to be cautious on this latter conclusion [that SOX2 levels correlate with OS and PFS]” (page 7, lines 218-219)
The last one, which assessed the stability of SOX2 expression from the first surgery to the recurrence, seems to be adequately supported by the experimental datas.
Finally, in the introduction and discussion, it would have been relevant to include reflection on the other molecular analyzes that participate or could participate in the classification of meningiomas.
Response: according to the Reviewer’s suggestion, we have added a reference to the molecular factors involved in meningioma classification, in the Introduction section, on page 2, lines 51-53, and in the Discussion section, on page 6, lines 183-190.
Overall an interesting and certainly relevant subject but the statements on the survival impact on SOX2 should be understated.
Response: We thank the Reviewer for having appreciated our work. In the revision, we have understated the survival impact of SOX2 expression in meningioma.
Reviewer 2 Report
Bonaventura et al. present a very interesting topic of the prediction power of SOX2 on aggressive intracranial meningiomas. They did a comprehensive analysis of patients with meningiomas with various levels of SOX2. It turns out SOX2 can serve as a strong prognostic factor both for overall and for recurrence-free survival, which makes it a valuable biomarker for meningiomas treatment.
In general, this study is properly designed, and the results are clear and sound. However, it might be helpful if including a section discussing the biological functions of SOX2, particularly its relevance to cancer progression. It is also good to see survival analysis using data from other source, such as TCGA, etc.
Author Response
Response to Reviewers’ comments
Reviewer #2
Bonaventura et al. present a very interesting topic of the prediction power of SOX2 on aggressive intracranial meningiomas. They did a comprehensive analysis of patients with meningiomas with various levels of SOX2. It turns out SOX2 can serve as a strong prognostic factor both for overall and for recurrence-free survival, which makes it a valuable biomarker for meningiomas treatment.
In general, this study is properly designed, and the results are clear and sound.
Response: we thank the Reviewer for having appreciated our work.
However, it might be helpful if including a section discussing the biological functions of SOX2, particularly its relevance to cancer progression.
Response: according to the Reviewer’s suggestion, we have expanded the section discussing SOX2 role in cancer progression, in the Discussion section, on pages 6-7, lines 193-207.
It is also good to see survival analysis using data from other source, such as TCGA, etc.
Response: the Reviewer raises an important point. Before starting our original work, we had tried to analyze available data. However, a database about meningioma is not present in the TCGA. In other datasets such as the EMBL's European Bioinformatics Institute (EMBL-EBI) one, in which array analysis of a series of 68 meningioma is reported, the clinical and the pathological parameters are scarce. In particular, the WHO grade of meningioma, the therapeutical treatment, the recurrence or progression statuses are not reported. Other smaller series carry the same limitations. For these reasons, we decided to study a selected, homogeneously treated cohort of meningioma patients.
Round 2
Reviewer 1 Report
The authors improved their manuscript and took into account the recommendations proposed in the first review. This type of study remains unsuitable for studying survival data but the rest of the scientific data remains valid.